# Investigating the Effectiveness of a Workplace Musculoskeletal Disorders Management Program

**DOI:** 10.3390/healthcare12181815

**Published:** 2024-09-10

**Authors:** Eleni Grana, Petros Galanis, Emmanuel Velonakis, Styliani Tziaferi, Panayota Sourtzi

**Affiliations:** 1Department of Nursing, National and Kapodistrian University of Athens, 11527 Athens, Greece; elinagrana@nurs.uoa.gr (E.G.); pegalan@nurs.uoa.gr (P.G.); evelonak@nurs.uoa.gr (E.V.); 2Department of Nursing, University of Peloponnese, 22100 Tripoli, Greece; stziafer@uop.gr

**Keywords:** MSDs, musculoskeletal disorders, workplace, intervention, absenteeism, presenteeism

## Abstract

Musculoskeletal disorders (MSDs) present a major occupational health challenge, especially among office workers. **Objective**: To evaluate the effectiveness of a workplace intervention program for managing MSDs and measure its impact on presenteeism/absenteeism and health status. **Methods**: This semi-experimental study was conducted between May 2019 and October 2022 in Greece and involved 247 office workers randomly assigned to early and late intervention groups. The intervention included a seminar on ergonomics and proper workstation practices with a demonstration of exercises. Data were collected by the SF-36, the Standardised Nordic, and the Work Productivity and Activity Impairment Questionnaires. **Results**: In the early intervention group (n = 143), presenteeism significantly decreased (*p* = 0.045), from a mean value of 0.11 to 0.07, but not absenteeism. Significant improvements were observed in physical (*p* = 0.007) and mental health (*p* = 0.012). Reductions in pain were recorded for the neck (*p* = 0.032), shoulders (*p* = 0.015), wrists (*p* = 0.014), upper back, lower back, hips, and knees (*p* = 0.044 for each). In the late intervention group (n=104), there were no significant changes in absenteeism or presenteeism. Mental health improved significantly (*p* = 0.008), and reductions in pain were noted for the neck (*p* = 0.001), wrist (*p* = 0.0005), and upper back (*p* = 0.001). **Conclusions**: This workplace intervention program proved effective both in reducing pain in various body areas and reducing presenteeism, and improved physical and mental health were observed in the intervention groups while absenteeism remained unchanged.

## 1. Introduction

Musculoskeletal disorders (MSDs) are a significant challenge for working population health as well as for the economy worldwide. MSDs are the most frequently occurring occupational diseases in the European Union (EU) and do not discriminate across sectors and occupations. Even though MSDs can be prevented, they remain the most common work-related health problem globally. This issue raises concerns not only due to the impacts on workers’ health but also due to the detrimental effects on businesses and national economies [1].

According to EU-OSHA (European Agency for Safety and Health at Work) [2]:Three out of five workers in the European Union experience musculoskeletal discomfort, while one in five workers experienced chronic pain and discomfort in the lower back or neck in the past year.The most common musculoskeletal complaints are pain in the lower back, upper limbs, and neck, while discomfort in the lower limbs is less frequent.Sixty percent (60%) of workers with more than one occupational disease report that musculoskeletal problems are the most burdensome.

Absenteeism due to MSDs accounts for a significant proportion of lost workdays in the EU member states. In 2015, 53% of workers with MSDs reported being absent from work in the past year, compared to 32% of healthy workers. Workers with MSDs are absent not only more frequently but also for longer periods; 26% of workers with MSDs and another illness reported being absent for more than eight days in the past year, compared to 7% of workers without health problems [2]. Furthermore, the occurrence of MSDs in the workforce is associated with a high rate of presenteeism, early retirement, and financial difficulties [3].

The annual prevalence of MSDs in the U.S. for workers using computers ranges from 33.8% to 95.3%, and the medical costs combined with the increased absenteeism cost employers and the state between 45 and 54 billion U. S. dollars annually. MSDs undoubtedly led to lost workdays (absenteeism), reduced productivity (presenteeism), early retirement, and unemployment. Workers with MSDs may experience reduced work capacity, leading to decreased earnings, reduced productivity, and increased compensation costs due to absenteeism [4].

In a recent study by Gill et al. [5], it is projected that the number of people suffering from an MSD could increase to over one billion by 2050, doubling the current prevalence. This increase can be attributed to the aging population and the potential post-pandemic effects of COVID-19. Liu et al. [6] reported that lower back pain accounts for over 42% of the total years lost due to disability from MSDs globally. This significant burden underscores the importance of creating and implementing targeted intervention programs for managing MSDs and alleviating pressure on healthcare systems and economies [6].

Research in the literature regarding the effectiveness of different kinds of interventions at the workplace indicates that a combination of methods, such as ergonomics seminars and the provision of ergonomic equipment (e.g., height-adjustable desks), significantly reduces total sitting time and pain associated with musculoskeletal disorders [7,8,9,10,11]. Strengthening exercises have been shown to specifically alleviate shoulder pain [12,13,14]. The most effective interventions focus on redesigning the work environment, education and increasing knowledge among workers, and adopting ergonomic work practices [15]. This holistic approach underscores the importance of integrating ergonomic training and equipment with physical exercise and comprehensive workplace redesign to mitigate the impact of MSDs.

Considering the seriousness of the problem and the absence of relevant studies in office workers in Greece, at the time this study was designed and implemented, we aimed (a) at recording the musculoskeletal discomfort experienced by employees who use computers daily and (b) to evaluate the effectiveness of a health promotion program designed to prevent and manage musculoskeletal pain. 

## 2. Materials and Methods

### 2.1. Study Design

We conducted a quasi-experimental study from May 2019–October 2022. We enrolled office workers that used a computer daily from different companies located in Athens, Greece. The 8 companies that participated in the study all belonged in the private sector, and only employees that were office employees with daily use of computers were included. Inclusion criteria were (a) office workers who (b) worked using a computer in a sitting position, (c) were adults, and (d) participated voluntarily. The questionnaires were anonymous, were in printed form, and were collected by conducting prearranged visits at the workplace. The study was conducted in three phases, as shown in Table 1. Initially, both the early intervention and late intervention groups completed the baseline questionnaires. The intervention was then administered to the early intervention group. One month later, both groups completed the questionnaires for the second time, after which the late intervention group received the intervention. Four months later, follow-up questionnaires were administered to both groups to assess long-term effects. In May 2019, employees of 3 companies entered the study, then there was an interruption due to the COVID-19 pandemic, and the rest entered the study in early 2022. The intervention was applied face-to-face in groups of employees in each company by the first author of this study. 

The health promotion program consisted of a tailored seminar focused on ergonomics and proper computer workstation practices, supplemented with videos and demonstrations of stretching exercises (see Appendix A). Each exercise was explained in detail, with short videos shown simultaneously to illustrate the correct technique. The program aimed to provide participating companies with the material to facilitate continuous employee training on ergonomics and the prevention or management of musculoskeletal discomfort. The training sessions lasted 45 to 60 min in total. The exercises combined stretching and muscle strengthening and took no more than 3 to 5 min to perform. They required no athletic equipment, special clothing, or dedicated space. Employees were encouraged to perform these exercises 2 to 4 times during their 8 h workday, particularly when experiencing physical fatigue from prolonged sitting. The exercises targeted muscle groups commonly affected by musculoskeletal discomfort. Τhe participants from each company were firstly divided into early and late intervention groups and then into subgroups, and each training was carried out in small groups of 15–20 employees, so that the production process was not observed.

### 2.2. Measurements

We used an anonymous questionnaire that collected demographic and professional characteristics. The study variables were measured with the following questionnaires, after obtaining the necessary licenses:the Standardised Nordic questionnaire for the analysis of musculoskeletal symptoms, to evaluate the prevalence of musculoskeletal pain;the SF36 Health Survey, to evaluate health status;the Work Productivity and Activity Impairment Questionnaire, to capture absenteeism and presenteeism due to health issues.

The Standardised Nordic questionnaire [16] is a self-administered scale that serves as a screening tool for musculoskeletal disorders and has been translated and standardised in Greek [17]. It includes the general NMQ (Nordic Musculoskeletal Questionnaire), which refers to the whole body divided into nine anatomical areas: neck, shoulder blade areas and shoulders, elbows, wrists and hands, upper back (thoracic area), lower back (lumbar area), hips, knees, and ankles. The questionnaire has been widely used in different languages and working populations, and it has been reported as a useful and reliable tool. 

The SF-36v2 [18] is among the most widely used tools for assessing health-related quality of life. The SF-36v2 Health Survey is a concise, comprehensive, and generic questionnaire. Its eight major dimensions of health are summed up in two sub-scales, overall physical and overall mental health, and it has been translated and validated in Greek [19]. 

The Work Productivity and Activity Impairment Questionnaire: General Health (WPAI-GH) was selected to measure absenteeism and presenteeism over the past seven days, both during work and outside of work [20]. It examines the impact of employees’ health problems, both physical and mental, on their ability to work and perform activities outside of work. WPAI has also been translated and validated in Greek nurses [21]. The internal consistency for non-specific patient populations has been found to have an acceptable Cronbach’s alpha coefficient (over 0.74). The following four main outcomes can be derived from the WPAI-GH and expressed as percentages by multiplying the scores by 100 [22]:Percentage of work time missed due to health, for those currently employed.Percentage of impairment while working due to health, for those currently employed and who worked in the past seven days.Overall work impairment due to health, for those currently employed.Percentage of activity impairment due to health, for all respondents. For those who lost their jobs and did not actually work in the past seven days, the percentage of overall work impairment due to health will be equal to the percentage of work time missed due to health.

### 2.3. Ethical Considerations

The study protocol was approved by the Ethics Committee of the Department of Nursing of the National and Kapodistrian University of Athens (No: 153/13.11.2015). We conducted this study in accordance with the Declaration of Helsinki. Individual data of office employees were not collected, and informed consent was obtained. Participation in the study was voluntary. Participants were provided with detailed written information about the study’s purpose and procedures and their rights, including guarantees of confidentiality and the ability to withdraw from the study at any point without consequence.

### 2.4. Statistical Analysis

Descriptive statistics (means and standard deviations) were calculated for continuous variables at each time point for both groups. We also present categorical variables with numbers and percentages. Paired-samples *t*-tests compared pre- and post-intervention mean scores within each group, while independent-samples *t*-tests compared mean scores between groups. McNemar’s test assessed changes in binary outcomes within groups. Repeated-measures ANOVA evaluated differences across the three time points (baseline, one month, four months) and between groups. Chi-square tests compared categorical variables between groups. The two groups were not different according to demographic and job characteristics. Thus, we did not perform multivariable analysis since there were no confounding factors. Statistical significance was set at *p* < 0.05. We used the IBM SPSS 21.0 (IBM Corp. Released 2021. IBM SPSS Statistics for Windows, Version 21.0. Armonk, NY, USA: IBM Corp.) for the analyses.

## 3. Results

### 3.1. Demographic and Job Characteristics

A total of 247 workers participated in the study, all of whom were volunteers over the age of 18. All of them worked as office workers in a sedentary position with long hours of daily computer use. The 247 participants were randomly separated to the early intervention group and the late intervention group. As expected, a percentage of participants did not respond to all the evaluations. Figure 1 provides a detailed timeline of the intervention program’s implementation for both groups, as well as the number of initial participants who responded to subsequent assessments.

Most of the participants in the early intervention group were male (57.3%), and in the late intervention group 51% were female. The mean age of participants in the early intervention group was 37.41 years and in the late intervention group 35.95 years. Comparing the demographic and job characteristics in the two groups, we did not find any statistically significant differences (Table 2 and Table 3).

### 3.2. Study Measures

#### 3.2.1. Absenteeism/Presenteeism

Regarding the early intervention group, there was no statistically significant impact of the health education program on absenteeism, and the value remained at the same level from the baseline to the late evaluation measurement. Presenteeism, however, showed a statistically significant decrease from the early evaluation to the late evaluation measurement, as the mean value dropped from 0.11 to 0.07 (*p* = 0.045). Similarly, the index “% overall limited work capacity due to health” was significantly affected from the early evaluation to the late evaluation measurement, with a decrease from 0.12 to 0.08 and a *p* = 0.048. Finally, the index “% limited activity due to health” significantly decreased from the baseline to the late evaluation measurement, as it dropped from 0.16 to 0.10 (*p* = 0.025) (Figure 2 and Appendix A).

There were no statistically significant changes regarding absenteeism and presenteeism in the late intervention group, as the values for “% of working time lost due to health” and “% of limited capacity during work due to health” remained at the same level from the baseline 1 to the late evaluation. No change was recorded in the overall index “% overall limited work capacity due to health”. Finally, the index “% of limited activity due to health” appeared to decrease, but the reduction was not statistically significant (Figure 3 and Appendix A).

#### 3.2.2. SF36

In the early *intervention group*, the “Physical Health” index increased significantly from the baseline to the early evaluation, during which the educational seminar took place, as well as from the baseline to the late evaluation measurement. Specifically, the mean value increased from 78.06 in the baseline to 80.96 in the early evaluation (*p* = 0.007) and then to 81.19 in the late evaluation. Additionally, the “Overall Mental Health” index showed a statistically significant increase from the baseline to the early evaluation, following the completion of the educational seminar, with the mean value rising from 67.33 to 68.79 (*p* = 0.012) (Table 4 and Appendix A).

In the late intervention group, there appeared to be an improvement in the overall physical health index between evaluations, although the change was not statistically significant. Interestingly, a statistically significant improvement was recorded in the overall mental health index from the 1st baseline to the 2nd baseline measurement, despite the fact that no intervention was made (*p* = 0.008) (Table 4 and Appendix A). 

#### 3.2.3. Self-Reported MSDs

Regarding the extent of the MSDs as measured by the NMQ, the following changes were observed overtime in both groups (Appendix A). 

In the early *intervention group* (Figure 4), neck pain over the last 12 months and over the last 7 days decreased across all measurements with a statistically significant difference. Pain in the scapular regions decreased for pain over the last 7 days and significantly decreased from the early evaluation to the late evaluation (*p* = 0.013). There were also statistically significant reductions in pain in the wrist joints, both from the early evaluation to the late evaluation and for pain over the last 7 days. Pain in the upper back (thoracic region) decreased significantly from the early evaluation to the late evaluation (*p* = 0.044), as well as pain over the last 7 days from the baseline to the late evaluation and from the early evaluation to the late evaluation. Regarding lower back pain (lumbar region), there were significant decreases from the early evaluation to the late evaluation for pain over the last 12 months and from the early evaluation to the late evaluation and from the baseline to the late evaluation measurements for pain over the last 7 days. Furthermore, hip pain decreased significantly from the baseline to the late evaluation for both pain over the last 12 months and pain over the last 7 days, with (*p* = 0.044) and (*p* = 0.032), respectively. Overall, there was a significant reduction in pain across most body regions, especially for pain recorded over the last 7 days (Figure 4 and Appendix A).

In the *late intervention group* (Figure 5 and Appendix A) neck pain increased from 60% to 63% from the baseline 1 to the baseline 2, where no intervention had taken place. After participation in the educational seminar, neck pain significantly decreased to 48%. Neck pain over the last 7 days decreased significantly both from the baseline 2 to the late evaluation and from the baseline 1 to the late evaluation. There was a statistically significant reduction in shoulder pain, which decreased from 54% in the baseline 1 to 46% in the late evaluation. Shoulder pain, both over the last 12 months and over the last 7 days, decreased significantly from the baseline 1 to the late evaluation and from the baseline 2 to the late evaluation. Elbow pain increased slightly from the baseline 1 to the baseline 2 and then decreased slightly in the late evaluation, without a statistically significant change. Wrist joint pain was statistically significantly reduced from the baseline 1 to the late evaluation as well as from the baseline 2 to the late evaluation, (*p* = 0.029) and (*p* = 0.005), respectively. Pain in the upper back decreased significantly from the baseline 1 to the late evaluation, from 21% to 15%. Similarly, lower back pain also decreased significantly from the baseline 1 to the late evaluation. Pain in the hip and ankle showed a reduction from the baseline 1 to the late evaluation without a statistically significant change. In contrast, knee pain increased between all evaluations, but the changes were not statistically significant.

MSDs pain was also evaluated post intervention in both groups by sex, and in the early intervention group there were not any significant statistical differences between male and females. In the late intervention group, there was a statistically significant difference between males and females for the neck area for pain over the last 7 days (*p* = 0.027) (Appendix A).

Finally, it is worth mentioning that even though the questionnaire did not include a specific question to determine whether the participants continued to perform the exercises, during the collection of post-intervention questionnaires, many participants voluntarily discussed the effects of the exercises and mentioned the frequency with which they continued to engage in them.

## 4. Discussion

MSDs are major health problems affecting most workers regardless of occupational activity. Office work with long hours interacting with a computer is becoming the norm and has been reported as a cause for increased MSD complaints in the working population. Typically, working hours in Greece are 8 h per day, 5 days per week. The total hours that the employees in this study were using a computer daily are the combination of working and personal use hours per day. 

In this study, office workers were assigned into early and late intervention groups and an ergonomic and exercise intervention was implemented. There were no significant differences between the early intervention and late intervention groups in any of the reported demographic and professional characteristics. The effectiveness or lack of effectiveness of the intervention was assessed by its effect on absenteeism/presenteeism, health status, and self-reported MSD complaints pre-, post-, and 4-months-post-intervention evaluations.

Absenteeism increased from the baseline to the early evaluation measurements and then returned to the initial level in the early intervention group. This increase following participation in the intervention program can be attributed to improved health awareness among employees, leading to better observation of their sick days, or to alleviating feelings of fear and insecurity about participating in the study, which may have previously led to inaccurate reporting of health-related absenteeism. The late intervention group exhibited a similar pattern with no statistically significant changes, indicating that the health education program did not significantly impact absenteeism. These findings align with those of other studies that included absenteeism as an outcome measure in various occupation groups such as health care workers [23] and office workers [24]. However, there are also opposing results reported by Soler-Font et al. [9].

Presenteeism showed a statistically significant reduction in the early intervention group at the late evaluation. This suggests that the employees became better informed about health and wellness issues and their impact on productivity and performance after the intervention. On the contrary, the late intervention group showed no change, indicating the limited effect of the intervention in the short term. Increased levels of presenteeism and absenteeism due to MSDs at the workplace have been reported by other studies [25,26]. However, presenteeism has not been studied as an intervention outcome to our knowledge so far.

Comparing the physical and mental health scores assessed via the SF-36 questionnaire, the intervention program had a statistically significant positive impact on both variables in the early intervention group. Specifically, the average score for “overall physical health” increased at the early evaluation (*p* = 0.007) and further improved at the late evaluation (*p* = 0.013), indicating sustained improvement after the intervention. The late intervention group also showed slight improvements across all measurement periods. These results are consistent with [11], who found that a workplace health program incorporating strengthening exercises led to significant improvements in physical health among slaughterhouse workers experiencing musculoskeletal pain, particularly in the intervention group. The “Overall Mental Health” index also showed improvement in the early intervention group (*p* = 0.012) from the baseline to the early evaluation. However, in the late evaluation a slight decline was observed, though the index was still better than the initial level, indicating a positive but not long-term effect of the intervention on mental health. The late intervention group showed a similar pattern with a statistically significant improvement (*p* = 0.008) followed by a slight decrease at the third measurement, suggesting that mental health levels were not significantly impacted in the long run. This may mean that the office workers were influenced positively by the prospect of the future intervention. In support of our findings, Esmaeilzadeh et al. [27] has also reported that an intervention group demonstrated enhancements in functional status and both physical health- and mental health-related quality of life. 

Regarding MSDs, neck pain over the last 12 months and over the last 7 days decreased significantly in the early intervention group, with a statistically significant reduction from the baseline to the late evaluation. This indicates a positive long-term effect of the intervention on reducing neck pain. The late intervention group also showed statistically significant decreases in all neck pain measurements, particularly from the baseline 2 to the late evaluation, after the intervention was offered to them. Similar results have been reported by other studies [13,28]. Upper back and lower back pain in the early intervention group decreased, highlighting the positive impact on back pain experience post-intervention. The late intervention group also showed a significant decrease in upper back pain from the baseline 2 to the late evaluation, further supporting the effectiveness of the intervention. Other studies have shown that interventions that combine various methods are effective in reducing MSDs [7,8,9,10,16]. 

Shoulder pain did not decrease significantly in our study, and this may have been because of the specific exercises used (stretching). Strengthening exercises have been shown to specifically alleviate shoulder pain [12,13,14,29]. Elbow and wrist pain in the early intervention group showed reductions between measurements, and in the late intervention group, wrist pain decreased significantly from the baseline 2 to the late evaluation when the intervention was offered. These results contrast with Shuai et al. [8], who did not observe significant improvement in elbow and hand pain following an intervention program for teachers.

Pain reduction in various body parts, following an intervention, is the main positive outcome reported by many studies [8,27,30]. Upper back, shoulders, and neck seem to be the body parts with immediate benefits from an intervention aimed at preventing or managing pain [31]. 

Similarly to our finding, that the combination of interventions had positive effects on reducing MSDs and presenteeism and increasing health status, Aegerter et al. [32] have reported that the duration of health education programs, including tailored exercise regimens, ergonomic interventions, and education on ergonomic principles, plays a crucial role in maintaining long-term reductions in musculoskeletal pain, particularly in the neck.

### Limitations of the Study

The present study faces several limitations that could have influenced the results and their interpretation. Although participants were randomly assigned into the early intervention and late intervention groups, practical challenges during the program’s implementation and in coordinating with the various companies made it difficult to adhere strictly to the study design. Additionally, in this quasi-experimental intervention study, which included both early intervention and late intervention groups, participants in the late intervention group were aware from the beginning that they would eventually profit from the intervention, and this could have influenced their responses. The lack of blinding, as neither the participants nor the researcher were unaware of group assignments, might have affected the study’s outcomes. 

As the sample was limited to private sector companies, the generalizability of the findings is limited. Further studies including public sector employees could give a better idea of the effectiveness of the intervention. In addition, the use of self-administered questionnaires, although they are well established, affects the interpretation of the results; this could be rectified in future studies by applying mixed methods and/or objective measurements. Furthermore, the completion of the questionnaires by the late intervention group, following the application of the health promotion program, took place after 4 months, and this makes it difficult to compare the results in the parallel groups. Finally, the requirement for participants to complete the questionnaires three times during the study may have led to the notable dropout rate, which may have compromised the statistical power and the interpretability of the results. 

A not-foreseen obstacle during the study was the COVID-19 pandemic; it forced the study team to interrupt the study during the quarantine, and this has affected its timely completion as well as the long-term response rates. Consequently, while the results of this study are important, they should be interpreted with caution, taking into consideration the above limitations.

## 5. Conclusions

The intervention program significantly reduced pain and discomfort in the musculoskeletal system, particularly in the neck, shoulders, back, lower back, elbows, hips, knees, and ankles. These benefits persisted over time, demonstrating the program’s long-term effectiveness. While absenteeism remained unchanged, presenteeism decreased in the early intervention group, indicating a positive impact on productivity. Participants also reported better physical and mental health, highlighting the importance of comprehensive health programs. Implementing best practices could further enhance the work environment and support work-life balance. Given the high prevalence of MSDs among office workers, effective health and ergonomic interventions are crucial for improving employee well-being and organizational productivity.

## Figures and Tables

**Figure 1 healthcare-12-01815-f001:**
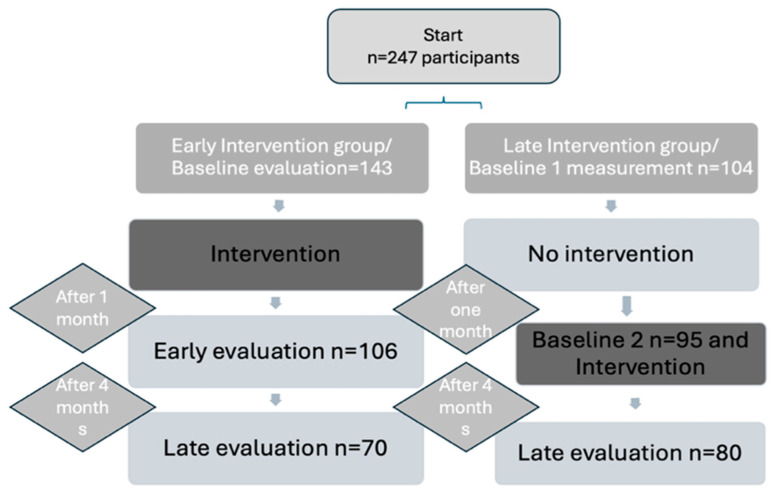
The time phases of the study in the two parallel groups and the number of participants that responded in every phase.

**Figure 2 healthcare-12-01815-f002:**
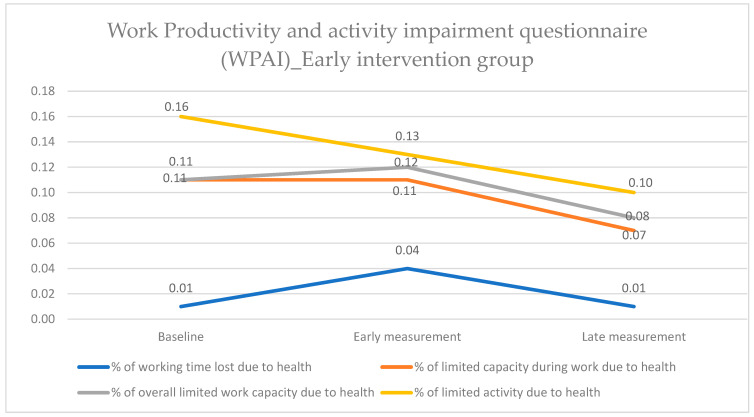
Work Productivity and Activity Impairment Questionnaire (WPAI). Comparison of the variables within the early intervention group at the three time points.

**Figure 3 healthcare-12-01815-f003:**
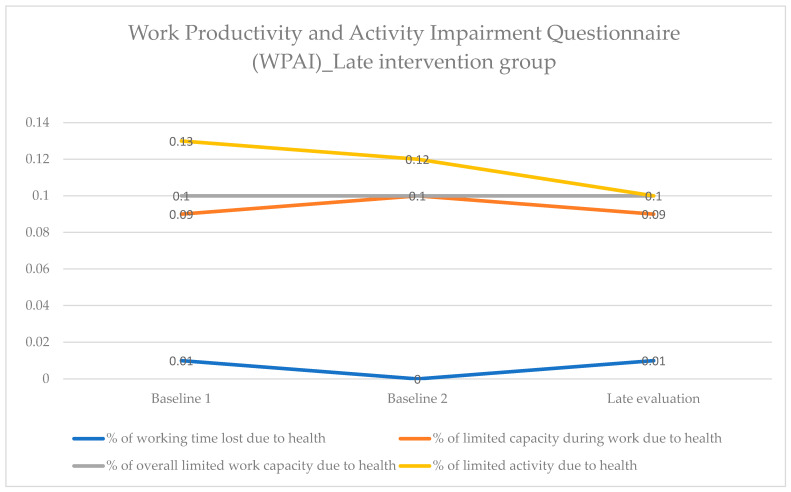
Work Productivity and Activity Impairment Questionnaire (WPAI). Comparison of the variables within the late intervention group at the three time points.

**Figure 4 healthcare-12-01815-f004:**
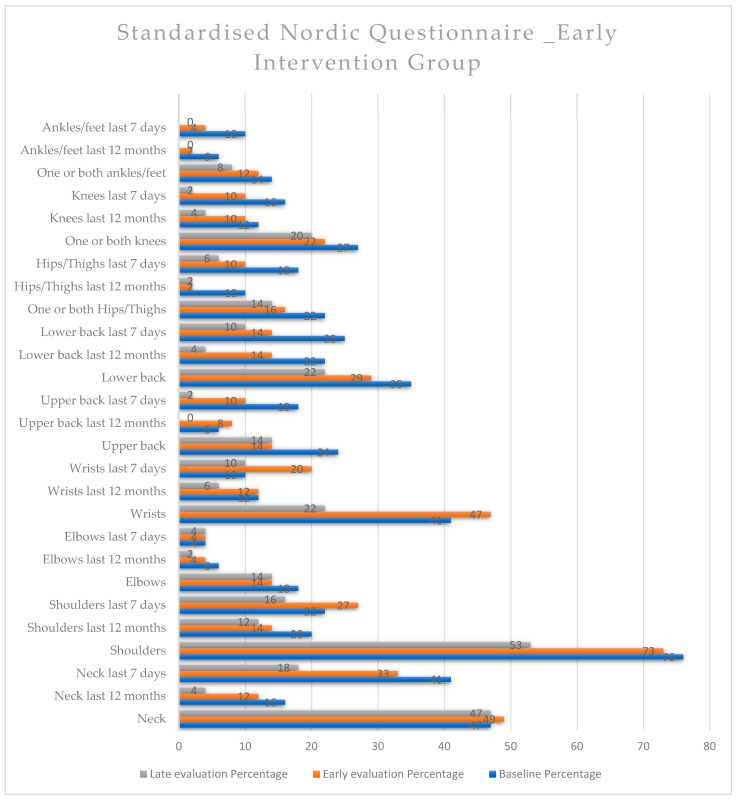
Standardised Nordic Questionnaire. Comparison of the variables within the early intervention group at the three time points.

**Figure 5 healthcare-12-01815-f005:**
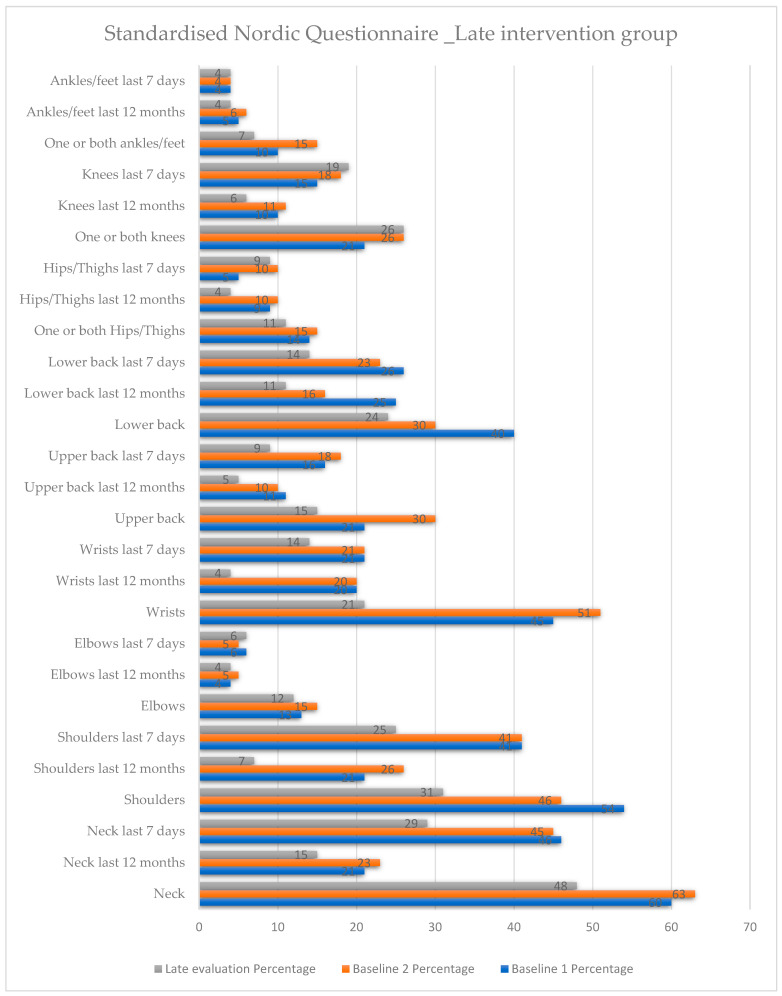
Standardised Nordic Questionnaire. Comparison of the variables within the late intervention group at the three time points.

**Table 1 healthcare-12-01815-t001:** The time phases of the study in the two parallel groups.

Time Phase	Phase 0	Phase 1	Phase 2 (One Month Post Intervention)	Phase 3 (4 Months Post Intervention)
Early intervention group	Baseline	Intervention	Early evaluation	Late evaluation
Late intervention group	Baseline 1	No intervention	Baseline 2 and intervention	Late evaluation

**Table 2 healthcare-12-01815-t002:** The demographic and job characteristics of n = 247 employees: Categorical variables.

Categorical Variables	Categories	Early Intervention Group (n = 143)		Late Intervention Group (n = 104)		*p*-Value ^a^
		Ν	%	Ν	%	
Gender	Male	82	57.3	51	49	0.196
	Female	61	42.7	53	51	
Marital status	Single	85	59.4	67	64.4	0.113
	Married	47	32.9	36	34.6	
	Divorced	9	6.3	1	1	
	Other	0		0		
Education level	High school graduate	15	10.5	8	7.7	0.313
	College/university graduate	77	53.8	66	63.5	
	Postgraduate/Doctorate	51	35.7	30	28.8	

^a^ Chi-square test.

**Table 3 healthcare-12-01815-t003:** The demographic and job characteristics of n = 247 employees: Continuous variables.

Continuous Variables Categories	Early Intervention Group (n = 143)		Late Intervention Group (n = 104)		*p*-Value ^c^
	Mean	Standard Deviation	Mean	Standard Deviation	
Age	37.41	9.722	35.95	9.243	0.235
Total years of work experience	13.32	9.234	11.88	8.579	0.212
Years of experience in the current position ^b^	9.07	8.164	8.33	10.335	0.529
Hours of computer use per day ^b^	9.58	2.183	9.64	1.723	0.805
Weekly working hours ^b^	42.92	5.58	42.32	4.158	0.351
Body weight ^b^	77.78	18.213	74.70	17.248	0.182
Height ^b^	174.1	9.196	172.38	9.178	0.147
BMI (Body Mass Index) ^b^	25.44	4.45	24.95	4.4	0.397

^b^ Mean value/standard deviation; ^c^ *t*-test.

**Table 4 healthcare-12-01815-t004:** SF-36 questionnaire. Comparison of variables between early and late intervention groups across three time points.

Groups		Baseline for Early Intervention Group/Baseline 1 for Late Intervention Group (Mean)	Early Evaluation for Intervention Group/Baseline 1 for Late Intervention Group (Mean)	Late Evaluation for Early and Late Intervention Groups (Mean)	*p*-Value between Baseline and Early Evaluation for Early Intervention Group/*p*-Value between Baseline 1 and Late Evaluation for Late Intervention Group ^a^	*p*-Value between Baseline and Late Evaluation for Early Intervention Group/*p*-Value between Baseline 1 and Late Evaluation for Late Intervention Group ^a^	*p*-Value between Early Evaluation and Late Evaluation for Early Intervention Group/*p*-Value between baseline 2 and Late Evaluation for Late Intervention Group ^a^
Early intervention group	Overall Physical Health	78.06	80.96	81.19	0.007	0.013	0.643
	Overall Mental Health	67.33	68.79	67.85	0.012	0.667	0.076
Late intervention group	Overall Physical Health	80.12	80.34	80.54	0.813	0.779	0.878
	Overall Mental Health	69.27	71.29	70.09	0.008	0.569	0.393

^a^ *t*-test for repeated measures.

## Data Availability

The original contributions presented in the study are included in the article, further inquiries can be directed to the corresponding author.

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
