# Peer review of "Investigating the Effectiveness of a Workplace Musculoskeletal Disorders Management Program"

_healthcare, 2024, doi:10.3390/healthcare12181815_

Round 1

Reviewer 1 Report

Comments and Suggestions for Authors

The manuscript is about a public health problem, musculoskeletal disorders, and aim to explore the effectiveness of a workplace intervention program (seminar and exercices) with a quasi experimental study. All participants received the intervention but at different time to allow a prolonged baseline in one group. 

Introduction - no comment to do

Method - The sentence "A total of 247...and control group." (lines 89-92) should be in results section. Information about companies approached is lacking (notion of private one is done in discussion section). Also, information about inclusion per person or cluster is not clear, because it seems difficult to perform an intervention in one by one. To clarify the timing of this study, notably measurement period, a timeline of parallel groups should be interesting. A description, or video in annexe, of the proposed exercices is warranted.  Could you precise the period of work (5 days per week or 6 days per week, because the mean weekly working hours is 42 with a 8-hour workday. Could you also precise why there are a mean of 9,5 hours per day of computer use (only work or also personal use) ? Could you explain if there is an immediate post intervention measure in "control" group after the intervention ? Like the 2 groups have an intervention, rename the group with "early" and "late" interventions rather than "intervention" and "control" should be more accurate. Precision about intervention date in the "control" group should be precise to correctly understand the results and discussion. Did you record observance to exercices at the last measurement ? 

Results - A flow chart should be useful to better described the inclusion, non inclusion, exclusion, lost of view... Results should be better understandable with figures rather than tables 3 to 8. Could you rename 1er, 2nd and 3rd measurement  : baseline, early evaluation, late evaluation for the "intervention group" and baseline 1, baseline 2 and late evaluation for the "control group" ?   The sentence "All participants... of the study" (lines 214-215) should be in discussion. The paragraph 3.2.3 is not understandable. Tables 7 & 8 are difficult to understand : "neck" is it reference to overall time in life ? If these tables are conserved, conception should be modified. 

Discussion - A verb is lacking in the first sentence (line 277). There is a double comma in line 284. There are probably letter inversion in lines 278 (intetacting), 303 (inverevtion), 304 (yo), 336 (showlder, stydy, anf), 349 (staturs), 355 (tandomly, interbvention), 356 (contro), 359 (conytol). Caution about interpretation are need, statistical significance is not minimal clinically important difference.  (ie doi: 10.1186/s12955-020-01344-w, DOI: 10.1016/j.spinee.2007.11.006 or DOI: 10.1016/j.jpain.2024.104559). Correction should be done regards to results comments. Is there any lost of view ?

Conclusion - If there is more than 2 months for last measure in "control" group, the conclusion is not supported by the results. 

Please, explain the acronyms EU-OSHA (line 37), NMQ (line 121) 

Abstract - should be improved and adapted, presented statistical data should be at the same measurement time. 

Author Response

Dear Reviewer 1,

Comment 1. Method - The sentence "A total of 247...and control group." (lines 89-92) should be in results section.

Answer 1. Thank you for the advice, we have corrected it.

Comment 2. Information about companies approached is lacking (notion of private one is done in discussion section).

Answer2. Information is completed in lines 85-86

Comment 3. Also, information about inclusion per person or cluster is not clear, because it seems difficult to perform an intervention in one by one.

 Answer3. Completed as advised in lines 97-98

Comment 4. To clarify the timing of this study, notably measurement period, a timeline of parallel groups should be interesting.

Answer 4. It is inserted, lines 95-97

Comment 5. A description, or video in annexe, of the proposed exercices is warranted.

Answer 5. Added as supplementary material

Comment 6.  Could you precise the period of work (5 days per week or 6 days per week, because the mean weekly working hours is 42 with a 8-hour workday. Could you also precise why there are a mean of 9,5 hours per day of computer use (only work or also personal use) ?

Answer 6. It’s the total hours of using a computer either for working or personal use per day. Completed as advised in discussion, lines 663-665.

Comment 7. Could you explain if there is an immediate post intervention measure in "control" group after the intervention ?

Answer 7. No, there was not immediate post intervetion measure in the control group. The measure was done 4 months post intervention. A figure showing the intervention/measurement phases has been added.

Comment 8. Like the 2 groups have an intervention, rename the group with "early" and "late" interventions rather than "intervention" and "control" should be more accurate.

Answer 8. We have changed the names of intervention group to early intervention group and control group to late intervention group as advised thank you.

Comment 9. Precision about intervention date in the "control" group should be precise to correctly understand the results and discussion.

Answer 9. The intervention was taking place after the second time the participants completed the questionnaires, usually 1 month after the 2nd measurement.

Comment 10. Did you record observance to exercices at the last measurement ?  

Answer 10. There was not a question in the questionnaire to clarify if the participants continued to perform the exercises, but when the researcher was collecting the post intervention questionnaires, most of them were discussing about the effect of the exercises and how often they perform them.

Comment 11. Results – A flow chart should be useful to better described the inclusion, non inclusion, exclusion, lost of view...

Answer 11. Flowchart added in the text

Comment 12. Results should be better understandable with figures rather than tables 3 to 8.

Answer 12. We have replaced tables with figures, except for the tables presenting SF36 findings, and the rest are included as supplementary material.

Comment 13. Could you rename 1er, 2nd and 3rd measurement : baseline, early evaluation, late evaluation for the "intervention group" and baseline 1, baseline 2 and late evaluation for the "control group" ?  

Answer 13. It is changed as advised.

Comment 14. The sentence "All participants... of the study" (lines 214-215) should be in discussion.

Answer 14. Done, pasted in discussion.  

Comment 15. The paragraph 3.2.3 is not understandable.

Answer 15. We have shortened the description of the findings as these are shown in tables and figures in detail.

Comment 16. Tables 7 & 8 are difficult to understand : "neck" is it reference to overall time in life ?

Answer 16. For the neck area, as an example, the questionnaire is asking firstly if the participant felt pain in the neck area for the last moths. Then there are 2 more specific questions, the 2nd question asks if the felt pain they felt the neck area for the last 12 months, was so intense that they couldn’t work or complete their tasks and the last question is if they felt pain in the neck area for the last 7 days.

Comment 17. If these tables are conserved, conception should be modified. 

Answer 17. We have replaced some of the tables with figures and we have uploaded supplementary tables and figures. Hopefully we have succeeded to make it easier to understand.

Comment 18. Discussion - A verb is lacking in the first sentence (line 277). There is a double comma in line 284. There are probably letter inversion in lines 278 (intetacting), 303 (inverevtion), 304 (yo), 336 (showlder, stydy, anf), 349 (staturs), 355 (tandomly, interbvention), 356 (contro), 359 (conytol).

Answer 18. All typing errors have been corrected.

Comment 19. Caution about interpretation are need, statistical significance is not minimal clinically important difference.  (ie doi: 10.1186/s12955-020-01344-w, DOI: 10.1016/j.spinee.2007.11.006 or DOI: 10.1016/j.jpain.2024.104559). Correction should be done regards to results comments.

Answer 19. Thank you for this important comment. We have included this caution in the results interpretation in the study limitations.

Comment 20. Is there any lost of view ?

Answer 20. Yes, they were; we have added the flow chart in the results section, where the participants that did not respond in the subsequent measurements are shown. In the analyses only respondents in the specific measurements have been used and shown in the results.

Comment 21. Conclusion - If there is more than 2 months for last measure in "control" group, the conclusion is not supported by the results. 

Answer 21. Thank you for the useful comment, we have discussed this in the limitations of the study section.

Comment 22. Please, explain the acronyms EU-OSHA (line 37), NMQ (line 121) 

Answer 22. It has been added as advised.

Reviewer 2 Report

Comments and Suggestions for Authors

This semi-experimental study assessed the effectiveness of a workplace intervention program on MSDs among 247 office workers in Greece using 3 different questionnaires after an intervention program that compared with control group,  i have some comments to be adressed.

Affiliations 2 and 3 are the same, so the numbering should be modified accordingly.

Were there any exclusion criteria?

I suggest providing a figure explaining each phase of the study, including both the intervention and control groups. This will enhance the presentation.

Table 2: Correct the word "experience" (it was mistakenly written as "experiebce").

Remove all small "b" and "c" notations in the tables, as they are unnecessary. The same applies to the "a" notation in Table 1.

Each test used is clear, as most readers understand when chi-square tests or t-tests are used. This comment applies to all tables.

When mentioning p-values, it's sufficient to say "p = number" (lines 183-187). This format should be applied consistently throughout the manuscript.

I suggest merging Tables 5 and 6.

The symbol "&" is not typically used in academic articles. Replace it with "and" or a comma, as in "Tables 7 and 8" (line 220).

What was the rationale for measuring p-values between each measurement? This could be misleading and difficult to interpret. I believe comparing only pre- and post-intervention measurements is sufficient. For example, in Table 3, the results show significant differences between the 2nd and 3rd measurements and no significance between the 1st and 2nd or the 1st and 3rd, making interpretation difficult. This issue is also present in Table 7.

There are many variables and p-values, making interpretation challenging.

The conclusion is very long and should be shortened to one paragraph.

participant consents should be mentioned also in text

text presentations:

english editing should be done as there are many spelling errors and this is not acceptable  for example in line 355  ( into interbvention  and contro groups)    ,,  p value should be referred as (p = number)   as i mentioned before

Comments on the Quality of English Language

many spelling errors, this is not acceptable to submit an article with these type of errors 

Author Response

Dear Reviewer 2,

Comment 1. Affiliations 2 and 3 are the same, so the numbering should be modified accordingly.

Answer 1. It is changed as advised.

Comment 2. Were there any exclusion criteria? The type of job, for example workers in sales, drivers, cleaning personnel etc.

Answer 2. All participants were expected to be office employees in order to be included to the study.

Comment 3. I suggest providing a figure explaining each phase of the study, including both the intervention and control groups. This will enhance the presentation.

Answer 3. It has been added in the methods section.

Comment 4. Table 2: Correct the word "experience" (it was mistakenly written as "experiebce").

Answer 4. All typing errors have been corrected.

Comment 5. Remove all small "b" and "c" notations in the tables, as they are unnecessary. The same applies to the "a" notation in Table 1.

Answer 5. It is changed as advised.

Comment 6. Each test used is clear, as most readers understand when chi-square tests or t-tests are used. This comment applies to all tables.

Answer 6. It is changed as advised.

Comment 7. When mentioning p-values, it's sufficient to say "p = number" (lines 183-187). This format should be applied consistently throughout the manuscript.

Answer 7.  It is changed as advised.

Comment 8. I suggest merging Tables 5 and 6.

Answer 8. Thank you we have merged them.

Comment 9. The symbol "&" is not typically used in academic articles. Replace it with "and" or a comma, as in "Tables 7 and 8" (line 220).

Answer 9. It is changed as advised.

Comment 10. What was the rationale for measuring p-values between each measurement? This could be misleading and difficult to interpret. I believe comparing only pre- and post-intervention measurements is sufficient. For example, in Table 3, the results show significant differences between the 2nd and 3rd measurements and no significance between the 1st and 2nd or the 1st and 3rd, making interpretation difficult. This issue is also present in Table 7.

Answer 10. We are sorry it was not clear and thank you for pointing it out. We compared participants from each group separately who responded in the questionnaire at both the first and second time points, then those who responded at both the first and third time points, and finally those who responded at both the second and third time points. We applied the same selection process both to early and late intervention groups. This procedure was followed for all variables. This was necessary because the participants in the 2nd and 3rd measurements were fewer as shown in the flow diagram included in the text now.

Comment 11. There are many variables and p-values, making interpretation challenging.

Answer 11. We the addition of Figures and supplementary material we hope this is clearer now.

Comment 12. The conclusion is very long and should be shortened to one paragraph.

Answer 12. Thank you, this has been shortened.

Comment 13. participant consents should be mentioned also in text.

Answer 13. This information was added as advised, in the 2.3 paragraph lines 161-164

Comment 14. text presentations: english editing should be done as there are many spelling errors and this is not acceptable  for example in line 355  ( into interbvention  and contro groups)    ,,  p value should be referred as (p = number)   as i mentioned before

Answer 14. all spelling mistakes have been corrected

Reviewer 3 Report

Comments and Suggestions for Authors

 The manuscript shows an intervention program in office workers in Greece.

I have several suggestions for you.

At first, I would like to emphasize the large sample recruited in the present study.

Write the intervention programs in the abstract

Line 89-92 are results.

The study design section should be reorganized, and a new procedure section should be established to clearly explain how the study was developed. Who participated in the research? How many researchers?

Were the exercises individual or group exercises? Were they performed during the exercises?

Do sex differences in group composition not affect the results (82 vs. 51)?

Line 220, 263,avoid first person

Line 303 “inverevtion” correct

Line 322 a period is missing after "al."

Line 329 “thie ntervention” correct

Line 334 “untervention.” correct

Line 336 “Showlder” and “stydy”, “anf”: correct

Line 345 “showlders” correct

Line 347 “in [31)” add the square bracket

Line 355 “tandomly assigned into interbvention 355 and contro groups” correct

Line 359 “conytol” correct

It would be interesting to add whether there were differences by sex after the intervention.

Review the journal's style guidelines for the reference section.

Author Response

Dear Reviewer 3,

Comment 1. Write the intervention programs in the abstract

Answer 1. It is added as advised.

Comment 2. Line 89-92 are results.

Answer 2. It is changed as advised.

Comment 3. The study design section should be reorganized, and a new procedure section should be established to clearly explain how the study was developed.

Answer 3. It is changed as advised. Please see figure 1

Comment 4. Who participated in the research?

Answer 4. Office employees from 8 different Greek companies, this information has been added in the methods section.

 Comment 5. How many researchers?

Answer 5. The intervention and data collection was performed by the first author (PhD candidate) as it is noted in the cover letter. This has also been added in the methods section.

Comment 6. Were the exercises individual or group exercises? Were they performed during the exercises?

Answer 6. The exercises were demonstrated by the researcher during the intervention program, while a video was showing the correct way of executing the exercises. The stretching exercises were grouped by body part/ muscle area for example exercises for the neck, for the upper back, for the lower back, for arms and for legs. All exercised could be performed at the desk area, with no need for extra space, equipment, athletic clothes etc. The total duration of the exercises suggested was 2-4 minutes and its participants could choose which areas of the body were more painful in order to perform the stretches.

Comment 7. Do sex differences in group composition not affect the results (82 vs. 51)?

Answer 7. Demographic/occupational characteristics were not found to affect the results.

Comment 8. Line 220, 263,avoid first person, Line 303 “inverevtion” correct, Line 322 a period is missing after "al.", Line 329 “thie ntervention” correct, Line 334 “untervention.” correct, Line 336 “Showlder” and “stydy”, “anf”: correct, Line 345 “showlders” correct, Line 347 “in [31)” add the square bracket, Line 355 “tandomly assigned into interbvention 355 and contro groups” correct, Line 359 “conytol” correct

Answer 8. All typing errors have been corrected as advised. 

Comment 9. It would be interesting to add whether there were differences by sex after the intervention.

Answer 9. As noted in the previous comments sex did not affect the findings.

Comment 10. Review the journal's style guidelines for the reference section.

Answer 10. We have carefully reviewed references.

Round 2

Reviewer 2 Report

Comments and Suggestions for Authors

Authors adresses all issues successfully

just for p values.  add space (p = n) not (p=0) without spacing

remove bold for word intervention in table 1

the subheading in table 4 is very long for presentation , i suggest to use abbreviation or shorten it

the conclusion is still long in my opinion (16 lines). >> i suggest to try again to decrease it to 7-8 lines 

i have no other comments to add 

Author Response

Dear Reviewer 2,

Comment 1. just for p values.  add space (p = n) not (p=0) without spacing 

Answer 1. Thank you for the advice, we have corrected it.

Commnet 2. remove bold for word intervention in table 1 

Answer 2. Thank you for the advice, we have corrected it.

Comment 3. the subheading in table 4 is very long for presentation , i suggest to use abbreviation or shorten it 

Answer 3. We have changed it to “Comparison of variables between early and late intervention groups across three time points.

Comment 4. the conclusion is still long in my opinion (16 lines). 

Answer 4. We have shortened as much as possible.

Reviewer 3 Report

Comments and Suggestions for Authors

Dear authors, 

Thank you for following my commentaries.

Cover letter information is not available for readers, for that reason this information must be added in the manuscript.

However, I still think that an analysis by sex is missing, in the same way that you affirm that it does not influence can be reflected in the manuscript with the corresponding statistical tests and with objective data that support this affirmation.

Author Response

Dear Reviewer 4,

Comment 1.  Cover letter information is not available for readers, for that reason this information must be added in the manuscript.

Answer 1. Thank you for your comment, we have added it in the method section, lines 96-97.

Comment 2. However, I still think that an analysis by sex is missing, in the same way that you affirm that it does not influence can be reflected in the manuscript with the corresponding statistical tests and with objective data that support this affirmation.

Answer 2. Thank you for your comment. We have added a statement in the results, lines 281-284. We also have added a supplementary table S5, to be included in the supplementary material.